# Validation of the accuracy of the FAST™ score for detecting patients with at-risk nonalcoholic steatohepatitis (NASH) in a North American cohort and comparison to other non-invasive algorithms

**Tinsay A. Woreta**[1]*, **Mark L. Van Natta**[2], **Mariana Lazo**[3], **Arunkumar Krishnan**[1], **Brent A. Neuschwander-Tetri**[4], **Rohit Loomba**[5], **Anna Mae Diehl**[6], **Manal F. Abdelmalek**[6], **Naga Chalasani**[7], **Samer Gawrieh**[7], **Srinivasan Dasarathy**[8], **Raj Vuppalanchi**[7], **Mohammad S. Siddiqui**[9], **Kris V. Kowdley**[10], **Arthur McCullough**[8], **Norah A. Terrault**[11], **Cynthia Behling**[5], **David E. Kleiner**[12], **Mark Fishbein**[13], **Paula Hertel**[14], **Laura A. Wilson**[2], **Emily P. Mitchell**[2], **Laura A. Miriel**[2], **Jeanne M. Clark**[1], **James Tonascia**[2], **Arun J. Sanyal**[9], **for the NASH Clinical Research Network**[¶]

**1** Johns Hopkins University School of Medicine, Baltimore, Maryland, United States of America, **2** Johns Hopkins University Bloomberg School of Public Health, Baltimore, Maryland, United States of America, **3** Drexel University Dornsife School of Public Health, Philadelphia, Pennsylvania, United States of America, **4** Saint Louis University, St. Louis, Missouri, United States of America, **5** University of California San Diego School of Medicine, San Diego, California, United States of America, **6** Duke University, Durham, North Carolina, United States of America, **7** Indiana University School of Medicine, Indianapolis, Indiana, United States of America, **8** Cleveland Clinic, Cleveland, Ohio, United States of America, **9** Virginia Commonwealth University School of Medicine, Richmond, Virginia, United States of America, **10** Liver Institute Northwest, Seattle, Washington, United States of America, **11** University of Southern California, Los Angeles, California, United States of America, **12** Laboratory of Pathology, National Cancer Institute, National Institutes of Health, Bethesda, Maryland, United States of America, **13** Department of Pediatrics, Feinberg Medical School of Northwestern University, Chicago, Illinois, United States of America, **14** Division of Gastroenterology, Hepatology and Nutrition, Texas Children's Hospital, Baylor College of Medicine, Houston, Texas, United States of America

¶ Membership of the NASH Clinical Research Network is provided in the Acknowledgments.
* tworeta1@jhmi.edu

## Abstract

### Background and aims

Management of patients with NASH who are at elevated risk of progressing to complications of cirrhosis (at-risk NASH) would be enhanced by an accurate, noninvasive diagnostic test. The new FAST™ score, a combination of FibroScan® parameters liver stiffness measurement (LSM) and controlled attenuation parameter (CAP) and aspartate aminotransferase (AST), has shown good diagnostic accuracy for at-risk NASH (area-under-the-Receiver-Operating-Characteristic [AUROC] = 0.80) in European cohorts. We aimed to validate the FAST™ score in a North American cohort and show how its diagnostic accuracy might vary by patient mix. We also compared the diagnostic performance of FAST™ to other non-invasive algorithms for the diagnosis of at-risk NASH.

**Data Availability Statement:** All relevant data are within the paper and its Supporting information files.

**Funding:** The Nonalcoholic Steatohepatitis Clinical Research Network (NASH CRN) is supported by the National Institute of Diabetes and Digestive and Kidney Diseases (NIDDK) (grants U01DK061713, U01DK061718, U01DK061728, U01DK061731, U01DK061732, U01DK061734, U01DK061737, U01DK061738, U01DK061730, U24DK061730). Additional support is received from the National Center for Advancing Translational Sciences (NCATS) (grants UL1TR000439, UL1TR000436, UL1TR000006, UL1TR000448, UL1TR000100, UL1TR000004, UL1TR000423, UL1TR002649). This research was supported in part by the Intramural Research Program of the NIH, National Cancer Institute. All FibroScan® 502 Touch systems were provided to the NASH CRN Investigators by Echosens™ through a Clinical Trial Agreement with the NIDDK. The funders had no role in study design, data collection and analysis, decision to publish, or preparation of the manuscript.

**Competing interests:** Dr. Manal Abdelmalek: None for this project. Dr. Abdelmalek has served as a consultant to Merck, Bristol Myers Squibb, Novartis, Novo Nordisk, Hanmi, TaiwanJ, Madrigal and NGM Bio. Her institution has received grant support from Allergan, Intercept, Boehringer-Ingelheim, Bristol Myers Squibb, Madrigal, Novo Nordisk, NGM Bio, Novartis, Viking, Hanmi, TARGET NASH, Celgene, and Genentech. She receives royalties from Elsevier and Up-to-Date Dr. Cynthia Behling: Dr. Behling is a consultant for ICON, COVANCE, Eli Lilly, Hologic, and Leica outside the submitted work. Dr. Naga Chalasani: There are none for this paper. For full disclosure, Dr. Chalasani has ongoing consulting activities (or had in preceding 12 months) with Abbvie, Madrigal, Zydus, Galectin, Boehringer-Ingelheim, Altimmune, and Foresite. These consulting activities are generally in the areas of nonalcoholic fatty liver disease and drug hepatotoxicity. Dr. Chalasani has equity in RestUp, a home health care provider agency. Dr. Chalasani receives research grant support from DSM and Exact Sciences where his institution receives the funding. Dr. Anna Mae Diehl: Dr. Diehl has consulted for Allergan, Alderya Therapeutics, Boehringer-Ingelheim, Celgene, Filcitrine, IBM Watson Health, Lumena, Merk, Novartis, Pfizer, Pliant, Roche, Quest Diagnostics, and twoXAR. She has research collaborations with Allergan, Boehringer-Ingelheim, Bristol Myers Squibb, Conatus, Exalenze, Galactin, Galmed, Genfit, Gilead, Hanmi, Hi La, Immuron, Intercept,

## Methods

We studied adults with biopsy-proven non-alcoholic fatty liver disease (NAFLD) from the multicenter NASH Clinical Research Network (CRN) Adult Database 2 (DB2) cohort study. At-risk-NASH was histologically defined as definite NASH with a NAFLD Activity Score (NAS) $\geq$ 4 with at least 1 point in each category and a fibrosis stage $\geq$ 2. We used the Echosens® formula for FAST™ from LSM (kPa), CAP (dB/m), and AST (U/L), and the FAST™-based Rule-Out (FAST™ $\leq$ 0.35, sensitivity = 90%) and Rule-In (FAST™ $\geq$ 0.67, specificity = 90%) zones. We determined the following diagnostic performance measures: AUROC, sensitivity (Se), specificity (Sp), positive predictive value (PPV), and negative predictive value (NPV); these were calculated for the total sample and by subgroups of patients and by FibroScan® exam features. We also compared the at-risk NASH diagnostic performance of FAST™ to other non-invasive algorithms: NAFLD fibrosis score (NFS), Fibrosis-4 (FIB-4) index, and AST to platelet ratio index (APRI).

## Results

The NASH CRN population of 585 patients was 62% female, 79% white, 14% Hispanic, and 73% obese; the mean age was 51 years. The mean (SD) AST and ALT were 50 (37) U/L and 66 (45) U/L, respectively. 214 (37%) had at-risk NASH. The AUROC of FAST™ for at-risk NASH in the NASH CRN study population was 0.81 (95% CI: 0.77, 0.84. Using FAST™-based cut-offs, 35% of patients were ruled-out with corresponding NPV = 0.90 and 27% of patients were ruled-in with corresponding PPV = 0.69. The diagnostic accuracy of FAST™ was higher in non-whites vs. whites (AUROC: 0.91 vs 0.78; p = 0.001), and in patients with a normal BMI vs. BMI > 35 kg/m$^2$ (AUROC: 0.94 vs 0.78, p = 0.008). No differences were observed by other patient characteristics or FibroScan® exam features. The FAST™ score had higher diagnostic accuracy than other non-invasive algorithms for the diagnosis of at-risk NASH (AUROC for NFS, FIB-4, and APRI 0.67, 0.73, 0.74, respectively).

## Conclusion

We validated the FAST™ score for the diagnosis of at-risk NASH in a large, multi-racial population in North America, with a prevalence of at-risk NASH of 37%. Diagnostic performance varies by subgroups of NASH patients defined by race and obesity. FAST™ performed better than other non-invasive algorithms for the diagnosis of at-risk NASH.

## Introduction

Nonalcoholic fatty liver disease (NAFLD) is the most common cause of liver disease worldwide and affects more than 25% of the global population [1]. It is strongly linked with obesity and the metabolic syndrome, a cluster of conditions that increases the risk of cardiovascular disease and includes abdominal obesity, insulin resistance, dyslipidemia, and hypertension. A subset of patients with NAFLD have nonalcoholic steatohepatitis (NASH) and are at higher risk of developing liver-related morbidity and mortality from progression to cirrhosis, hepatocellular carcinoma, or need for liver transplantation.

Madrigal Metabolomics, NGM Pharmaceuticals, Prometheus, and Shire. Dr. Samer Gawrieh: Dr. Gawrieh has consulted for TransMedics and Pfizer and received research grant support from Cirius, Galmed, Viking, Zydus and Sonic Incytes. Dr. Kris Kowdley: Dr. Kowdley has consulted for Akero, Calliditas, Corcept, CymaBay, Enanta, Genfit, Gilead, HighTide, Inipharm and Intercept. His institution has received grant and research support from Allergan, Enanta, Galectin, Gilead, Immuron, Intercept, Novartis, Prometheus, and Zydus. Dr. Kowdley is on the Advisory Board for Conatus and Gilead, and is on the Speaker Bureau for Abbvie, Gilead and Intercept. Dr. Rohit Loomba: There are none for this paper. Dr. Loomba serves as a consultant or advisory board member for 89bio, Alnylam, Arrowhead Pharmaceuticals, AstraZeneca, Boehringer Ingelheim, Bristol-Myer Squibb, Cirius, CohBar, DiCerna, Galmed, Gilead, Glympse bio, Intercept, Ionis, Metacrine, NGM Biopharmaceuticals, Novo Nordisk, Pfizer, Sagimet and Viking Therapeutics. In addition, his institution has received grant support from Allergan, Boehringer-Ingelheim, Bristol-Myers Squibb, Eli Lilly and Company, Galmed Pharmaceuticals, Genfit, Gilead, Intercept, Inventiva, Janssen, Madrigal Pharmaceuticals, NGM Biopharmaceuticals, Novartis, Pfizer, pH Pharma, and Siemens. He is also co-founder of Liponexus, Inc. Dr. Loomba receives funding support from NIEHS (5P42ES010337), NCATS (5UL1TR001442), NIDDK (R01DK106419, 1R01DK121378, R01 DK124318, P30DK120515), and DOD PRCRP (CA170674P2). Dr. Brent Neuschwander-Tetri: None for the project. Consultant or advisor for Akero, Alimentiv, Allergan, Alnylam, Amgen, Arrowhead, Axcella, Boehringer Ingelheim, BMS, Durect, Enanta, Fortress, Gelesis, Genfit, Gilead, HepGene, High Tide, HistoIndex, Intercept, Ionis, LG Chem, Lipocine, Madrigal, Medimmune, Merck, Mirum, NGM, NovoNordisk, pH-Pharma, Sagimet, Siemens, Theratechnologies, 89Bio; Institutional research grants: Allergan, BMS, Cirius, Enanta, Genfit, Gilead, Intercept, Madrigal, NGM Dr. Arun Sanyal: None for this project. Dr. Sanyal is President of Sanyal Biotechnology and has stock options in Genfit, Akarna, Tiziana, Indalo, Durect Inversago and Galmed. He has served as a consultant to Astra Zeneca, Nitto Denko, Conatus, Nimbus, Salix, Tobira, Takeda, Jannsen, Gilead, Terns, Birdrock, Merck, Valeant, Boehringer-Ingelheim, Bristol Myers Squibb, Lilly, Hemoshear, Zafgen, Novartis, Novo Nordisk, Pfizer, Exhalenz and Genfit. He has been an unpaid consultant to Intercept, Echosens, Immuron, Galectin, Fractyl, Syntlogic, Affimune, Chemomab, Zydus, Nordic

Due to the growing pandemic of obesity, the prevalence of NASH is increasing, and NASH is projected to become the leading cause of end-stage liver disease and liver transplantation in the upcoming decades [2]. Studies have shown that compared with the general population, patients with NAFLD have higher overall and liver-related mortality [3]. NASH related mortality is 1.7 times higher than in NAFLD (26 vs. 15 events per 1000 person-years), and liver-specific mortality is 15 times higher than in NAFLD (12 vs. 0.8 events per 1000 person-years) [1]. Currently, there are no FDA approved pharmacologic therapies for NASH. There are multiple ongoing Phase 3 clinical trials investigating the efficacy of promising novel agents with anti-inflammatory or anti-fibrotic properties for the treatment of NASH [4].

An accurate assessment of fibrosis stage is critical in the evaluation of patients with NASH, as studies have shown that the risk of liver-related mortality is directly related to fibrosis stage [5]. The presence and severity of fibrosis is an independent predictor for long-term prognosis in patients with NAFLD, including overall mortality [6].

Liver biopsy remains the gold standard for the detection of steatohepatitis and staging of fibrosis for patients with NASH [7]. However, it is an imperfect standard due to its invasive nature, the potential for rare but life-threatening complications, high cost, sampling error, and inter-/intra-observer variability. Furthermore, in clinical practice, patients are often reluctant to undergo a liver biopsy due to a fear of an invasive procedure and the risk of complications. These limitations have fueled researchers to develop alternative noninvasive strategies for fibrosis assessment, and in recent years it has become an area of intense research [8].

Currently, there are a number of commonly used noninvasive tools for the detection of advanced fibrosis which have been validated in patients with NASH. These include scoring algorithms that can be easily calculated using readily available clinical and laboratory data, serum biomarkers, and imaging modalities [9]. The most widely used scoring systems are the NAFLD fibrosis score (NFS) (based on age, BMI, presence of impaired fasting glucose or diabetes, AST/ALT ratio, platelet count, and albumin), FIB-4 index (based on age, AST, ALT, and platelet count), and AST to platelet ratio index (APRI) [7]. Vibration controlled transient elastography (FibroScan®), which measures liver stiffness using an ultrasound-based technology, was approved by the U.S. FDA in 2013 for use as an aid to the clinical management of patients with liver disease [10,11]. It is an inexpensive, easy to perform test that has been shown to have high diagnostic accuracy for the detection of advanced fibrosis in patients with NASH [12]. FibroScan® can also provide a quantitative assessment of steatosis using the controlled attenuation parameter (CAP) [13]. FibroScan® is equipped with two probes for adults: the standard M probe and the extra-large (XL) probe for obese patients.

The main limitations of these currently used noninvasive tools are: (1) although they have high diagnostic accuracy in the detection of advanced fibrosis (i.e., bridging fibrosis or cirrhosis) [12,14], they are less accurate for identifying fibrosis stage 2 or above [7], and (2) they do not detect the presence or degree of steatohepatitis and other markers of liver cell injury such as hepatocellular ballooning. There is thus a great need to identify noninvasive diagnostic tools that can predict the presence of inflammation in addition to fibrosis and identify individuals who are at the highest risk of disease progression. In particular, it would be very beneficial to have a highly accurate, noninvasive diagnostic test to identify patients with NASH with stage 2 fibrosis or higher and more advanced grades of steatohepatitis and hepatocellular injury (NAS ≥ 4), as such patients are at elevated risk of fibrosis progression and the development of cirrhosis.

Recently, the FibroScan-AST (FAST™) score was reported as a new and promising noninvasive tool to identify patients with progressive NASH [15]. It is derived from a model

Bioscience, Albireo, Prosciento, Surrozen. His institution has received grant support from Gilead, Salix, Tobira, Bristol Myers, Shire, Intercept, Merck, Astra Zeneca, Malinckrodt, Cumberland and Novartis. He receives royalties from Elsevier and UptoDate. Dr. Mohammad Siddiqui: Dr. Siddiqui is a consultant for Pfizer and is on an advising board for AMRA. Dr. Norah Terrault: Dr. Terrault received institutional grant support from Gilead Sciences, Roche-Genentech and GSK, and consulting for Gilead Sciences, Saol Therapeutics and Moderna. Dr. Raj Vuppalanchi: Dr. Vuppalanchi received a one-time consultant fee from EchoSens on spleen stiffness measurement. No conflicts of interest: Jeanne Clark, Srinivasan Dasarathy, Mark Fishbein, Paula Hertel, David Kleiner, Arunkumar Krishnan, Mariana Lazo, Arthur McCullough, Laura Miriel, Emily Mitchell, James Tonascia, Mark Van Natta, Laura Wilson, Tinsay Woreta This does not alter our adherence to PLOS ONE policies on sharing data and materials.

**Abbreviations:** AST, Aspartate aminotransferase; BMI, Body Mass Index; CAP, Controlled Attenuation Parameter; FAST, FibroScan AST score; LSM, Liver Stiffness Measure; NAFLD, Nonalcoholic Fatty Liver Disease; NAS, NAFLD Activity Score; NASH, Nonalcoholic Steatohepatitis.

combining liver stiffness measurement (LSM), CAP, and AST according to the following equation:

$$\text{FAST}^{\text{TM}} = \frac{e^{-1\cdot65+1\cdot07\times\ln(\text{LSM})+2\cdot66*10^{-8}\times\text{CAP}^3-63\cdot3\times\text{AST}^{-1}}}{1+e^{-1\cdot65+1\cdot07\times\ln(\text{LSM})+2\cdot66*10^{-8}\times\text{CAP}^3-63\cdot3\times\text{AST}^{-1}}}$$

It was found to be the most predictive model for the identification of patients with NASH with significant activity (NAFLD activity score (NAS) ≥ 4) and fibrosis (stage 2 or higher) in a cohort of 350 adult patients with suspected NAFLD attending liver clinics in England [15]. The diagnostic accuracy in this derivation cohort was good, with an area-under-the-Receiver-Operating-Characteristic (AUROC) of 0.80. Two thresholds for the FAST™ score were established using this derivation cohort, a rule-out cutoff of ≤ 0.35 to yield a sensitivity of 90% and a rule-in cutoff of ≥ 0.67 to yield a specificity of 90% [15]. The investigators assessed the accuracy of FAST™ using external validation cohorts comprised of seven independent international cohorts of patients with histologically confirmed NAFLD from France, China, Malaysia, Turkey, and the U.S. and found good performance with AUROC values ranging from 0.74 to 0.95 [15].

In this study, we sought to validate the FAST™ score in a large, diverse cohort of U.S. patients enrolled in the NASH Clinical Research Network (NASH CRN) Adult Database (D2) cohort study. Patients enrolled in this observational study had the full spectrum of NAFLD, a common context of use of FAST™ in patients found to have NAFLD by imaging. We also aimed to determine how the diagnostic accuracy of FAST™ varies across subgroups defined by various patient characteristics and FibroScan® exam features. Finally, we compared the diagnostic performance of FAST™ to its individuals components as well as other non-invasive algorithms and prediction models based on combinations of liver biochemistries.

## Methods

### Study design and participants

The validation study cohort was comprised of individuals 18 years or older with biopsy-proven NAFLD enrolled in the NAFLD Adult DB2 study conducted by the NASH CRN, a multicenter network sponsored by the National Institute of Diabetes and Digestive and Kidney Diseases (NIDDK).

The NAFLD Adult DB2 study is a prospective observational study of patients with biopsy-proven NAFLD which was started in December 2009 at 8 U.S. medical centers: Case Western Reserve University and Cleveland Clinic Foundation (Cleveland, OH); Duke University (Durham, NC); Indiana University (Indianapolis, IN); Saint Louis University (St. Louis, MO); University of California, San Diego (San Diego, CA); University of California, San Francisco (San Francisco, CA); Virginia Mason Medical Center and Swedish Medical Center (Seattle, WA); and Virginia Commonwealth University (Richmond, VA) [16]. The inclusion and exclusion criteria for the DB2 study were similar to that of the earlier Adult Database 1 study described in an earlier publication [17], with the exception that all new participants required histological evidence of NAFLD to be eligible for enrollment [16]. FibroScan® exam was performed for all partipants enrolled in the Adult DB2 database if they provided signed informed consent to participate. Individuals enrolled in the DB2 study were included in this analysis if they met the following criteria (i) had FibroScan® performed with simultaneous assessment of LSM and CAP and aminotransferase level measurements within 6 months of liver biopsy and (ii) had liver biopsy reviewed centrally by the NASH CRN Pathology Committee. The study was approved by the institutional review boards of participating institutions. All participants provided written informed consent prior to enrollment. Data were stored, monitored, and

analyzed at the Data Coordinating Center at the Johns Hopkins Bloomberg School of Public Health.

## Characterization of study participants and procedures

Detailed demographic, anthropometric, clinical, and laboratory data were systematically collected on all participants as part of the individual study protocol. Routine laboratory tests were performed on fresh blood samples in Clinical Laboratory Improvement Amendments-certified laboratories at each participating clinical site according to standard clinical protocols [17]. Liver biopsies were evaluated and scored centrally by the NASH CRN Pathology Committee using the NASH CRN validated histologic scoring system, which specifies the NAS and fibrosis stage [18]. A diagnosis of definite NASH was made by the Pathology Committee based on pattern recognition independent of NAS score [18]. LSM and CAP were simultaneously assessed using the FibroScan® 502 Touch device equipped with both M and XL probes (Echosens, Paris, France). The FibroScan® 502 Touch device was available at each participating clinical site. Probe selection was performed using the automatic probe selection tool embedded within the FibroScan® 502 Touch operating software, which is based on the probe to liver capsule distance.

All FibroScan® procedures were performed by study personnel who were trained and certified by Echosens. Individuals were asked to fast for at least 3 hours prior to the FibroScan® procedure. Participants were placed in the supine position with their right arm fully abducted. Ten valid measurements were obtained [16]. The final CAP and LSM measurements were recorded as the median values of 10 consecutive valid measurements, and they were expressed in dB/m and kPa, respectively. Unreliable LSM values were defined as medians with the inter-quartile range (IQR)/median > 30% [16].

At-risk NASH was defined as definite NASH on biopsy with a NAS 4 or higher with at least 1 point in each category and a fibrosis stage 2 or higher. The FAST™ score was calculated from liver stiffness (E, kPa), steatosis (controlled attenuation parameter [CAP], dB/m), and AST (U/L) according to the equation detailed above.

## Statistical analysis

Descriptive statistics including means, medians, standard deviations and inter-quartile ranges were used to determine the characteristics of the study cohort. Comparisons between patients with at-risk NASH and those without at-risk NASH were performed using chi-square tests for categorical variables and the unequal variance t-test for continuous data.

Specifying the rule-out cutoff of 0.35 and a rule-in cutoff of 0.67, diagnostic performance measures of the FAST™ score for the identification of at-risk NASH were computed using our validation study cohort. Sensitivity, specificity, positive predictive value (PPV), and negative predictive value (NPV) were calculated. PPV and NPV were calculated at varying prevalences using observed sensitivity and specificity. Discrimination (the ability of the FAST™ score to correctly classify those with and without at-risk NASH) was assessed using area under receiver operating characteristic (AUROC) curves. The Delong test was used to compare ROC curves [19]. The diagnostic performance measures of the FAST™ score for at-risk NASH were determined for subgroups of patients based on variables including demographics, anthropometrics, clinical characteristics, and FibroScan® exam features.

Various diagnostic models for the identification of at-risk NASH were compared to the FAST™ score [19]. In particular, the diagnostic performance of FAST™ was compared to the NAFLD fibrosis score (NFS), Fibrosis-4 (FIB-4) index, and AST to platelet ratio index (APRI).

Statistical analyses were performed using SAS 9.4 (2017 SAS Institute Inc., Cary, NC) and Stata 15.1 (StataCorp. 2017. Stata Statistical Software: Release 15. College Station, TX: Stata-Corp LLC.). P-values were two-sided, nominal and 0.05 was the threshold for statistical significance.

## Results

### Characteristics of the study population

A total of 585 adults with biopsy-confirmed NAFLD from the Adult DB2 study were included in the analysis. The baseline characteristics of our study cohort are depicted in Table 1. Overall, the mean age was 51 years. The majority of participants were white (79%), female (62%), and obese (73%). 41% had a body mass index (BMI) $\geq$ 40 kg/m$^2$. 4% were black, 9% were Asian, and 1% were American Indian/Pacific Islander. 14% were Hispanic. Type 2 diabetes mellitus (43%) and hyperlipidemia (49%) were common underlying comorbidities. The mean (SD) AST and ALT were 50 (37) U/L and 66 (45) U/L, respectively. The mean (SD) LSM was 11.4 (11.1) kPa, and CAP was 321 (52) dB/m. The XL probe was used for 324 patients (55%).

214 participants (37%) had at-risk NASH. Participants with at-risk NASH were more likely to be white (83% vs. 76%), female (71% vs. 57%), and obese (82% vs. 68%) when compared to those without at-risk NASH (Table 1). They were also more likely to be older (mean age 54 vs. 51 years) and have Type 2 diabetes mellitus (55% vs. 36%). At-risk NASH patients had higher mean AST (65 vs. 40 U/L), ALT (78 vs. 58 U/L), LSM (15.2 vs. 9.3 kPa), and CAP scores (327 vs. 317 dB/m) and lower platelet counts (227K vs. 243K cells/μL).

There were 61 patients (10%) with cirrhosis in the cohort, of whom 35 (57%) met criteria for at-risk NASH. 121 (21%) patients had stage 3 fibrosis, 83% of whom had at-risk NASH; 120 (21%) had Stage 2 fibrosis, 65% of whom had at-risk NASH (S1 Table).

### Diagnostic performance of FAST™ score

The AUROC of FAST™ for at-risk NASH in the study population was 0.81 (95% CI: 0.77, 0.84) (Fig 1).

35% of patients in the NASH CRN cohort had a FAST™ score below the rule-out cutoff of 0.35, and 27% had a score above the rule-in cutoff of 0.67. 38% of patients had scores in the interderminate range. Fig 1 shows a graphical display of the frequency distribution of FAST™ scores according to the presence or absence of at-risk NASH. FAST™ scores ranged from 0.01 to 0.96.

The diagnostic performance of the FAST™ score, i.e., the sensitivity, specificity, positive predictive value (PPV), and negative predictive value (NPV) using a rule-out cutoff of $\leq$ 0.35 and a rule-in cutoff of $\geq$ 0.67, is depicted in Table 2. The rule-out zone cutoff of 0.35 yielded a sensitivity of 0.91, specificity of 0.50, PPV of 0.51, and NPV of 0.90. The rule-in zone cutoff of 0.67 led to a sensitivity of 0.51, specificity of 0.87, PPV of 0.69, and NPV of 0.76.

Table 3 displays the diagnostic accuracy of the FAST™ score at varying prevalences of at-risk NASH, which highlights that PPV decreases with decreasing prevalence of at-risk NASH, while NPV increases.

**Subgroup analysis.** The diagnostic accuracy of the FAST™ score for at-risk NASH among various subgroups characterized by important patient characteristics including age, gender, race/ethnicity, and BMI and FibroScan® probe type is displayed in Table 4. The performance of FAST™ was better in non-whites vs. whites (AUROC: 0.91 vs 0.78; p = 0.001). When dividing the population into three BMI categories, the diagnostic accuracy of FAST™ did not differ significantly (Table 4). As a sensitivity analysis, when BMI was categorized into four groups, defining an additional group as BMI <25 kg/m$^2$, the AUROC for the 23 participants in the

**Table 1. Characteristics of the population by NASH activity and fibrosis status.**

| | | At-risk NASH† | | | P-value* |
|---|---|---|---|---|---|
| | | No<br>(n = 371) | Yes<br>(n = 214) | Total<br>(n = 585) | |
| | | Mean (SD)/n (%) | Mean (SD)/n (%) | Mean (SD) / n (%) | |
| **Demographics** | | | | | |
| Sex | Male | 161 (43.4) | 63 (29.4) | 224 (38.3) | 0.0008 |
| Race | White | 282 (76.0) | 178 (83.2) | 460 (78.6) | 0.04 |
| | Black | 11 (3.0) | 11 (5.1) | 22 (3.8) | |
| | Asian | 42 (11.3) | 12 (5.6) | 54 (9.2) | |
| | Am Indian/ Pacific Islander | 5 (1.4) | 3 (1.4) | 8 (1.4) | |
| | Refused | 27 (7.3) | 9 (4.2) | 36 (6.2) | |
| Ethnicity | Hispanic | 63 (17.0) | 19 (8.9) | 82(14.0) | 0.006 |
| Age | Years | 50 (12) | 54 (12) | 51 (12) | <0.0001 |
| **Anthropometrics** | | | | | |
| BMI (kg/m$^2$) | 18.5–24.9 | 17 (4.6) | 6 (2.8) | 23 (3.9) | 0.004 |
| | 25.0–29.9 | 100 (27.0) | 33 (15.4) | 133 (22.7) | |
| | 30–39.9 | 116 (31.3) | 72 (33.6) | 188 (32.1) | |
| | $\geq$ 40 | 138 (37.2) | 103 (48.1) | 241 (41.2) | |
| **Comorbidities** | | | | | |
| Diabetes | None | 235 (63.3) | 93 (43.5) | 328 (56.1) | <0.0001 |
| | Type 1 | 2 (0.5) | 3 (1.4) | 5 (0.8) | |
| | Type 2 | 134 (36.1) | 118 (55.1) | 252 (43.1) | |
| Hyperlipidemia | Yes | 174 (46.9) | 115 (53.7) | 289 (49.4) | 0.11 |
| **Histology** | | | | | |
| Time since bx | Days | 75 (36) | 82 (34) | 77 (36) | 0.02 |
| Fibrosis | Stage 0 | 142 (38.3) | 0 (0.0) | 142 (24.3) | <0.0001 |
| | Stage 1 | 140 (37.8) | 0 (0.0) | 140 (24.0) | |
| | Stage 2 | 42 (11.4) | 78 (36.4) | 120 (20.6) | |
| | Stage 3 | 20 (5.4) | 101 (37.2) | 121 (20.7) | |
| | Stage 4 | 26 (7.0) | 35 (16.4) | 61 (10.4) | |
| **Liver tests** | | | | | |
| AST | U/L | 40 (31) | 65 (41) | 50 (37) | <0.0001 |
| ALT | U/L | 58 (41) | 78 (50) | 66 (45) | <0.0001 |
| AST/ALT ratio | | 0.78 (0.32) | 0.91 (0.33) | 0.83 (0.33) | <0.0001 |
| GGT | U/L | 59 (74) | 95 (96) | 72 (85) | <0.0001 |
| Alkaline phos | U/L | 78 (27) | 90 (33) | 82 (30) | <0.0001 |
| Total bilirubin | mg/dL | 0.63 (0.38) | 0.63 (0.35) | 0.63 (0.37) | 0.95 |
| Direct bilirubin | mg/dL | 0.19 (0.10) | 0.18 (0.10) | 0.19 (0.10) | 0.38 |
| Albumin | g/dL | 4.41 (0.34) | 4.35 (0.37) | 4.39 (0.35) | 0.05 |
| Total protein | g/dL | 7.39 (0.45) | 7.44 (0.55) | 7.41 (0.49) | 0.22 |
| **Labs** | | | | | |
| INR | | 1.03 (0.08) | 1.06 (0.09) | 1.04 (0.09) | 0.0002 |
| Platelets | 1000/μL | 243 (76) | 227 (71) | 237 (74) | 0.01 |
| Glucose | mg/dL | 110 (35) | 121 (39) | 114 (37) | 0.0006 |
| **FibroScan®** | | | | | |
| LSM** | kPa | 9.3 (9.6) | 15.2 (12.4) | 11.4 (11.1) | <0.0001 |
| CAP** | dB/m | 317 (51) | 327 (51) | 321 (52) | 0.02 |
| Probe type | M | 181 (48.8) | 80 (37.4) | 261 (44.6) | 0.008 |

(*Continued*)

**Table 1.** (Continued)

| | | At-risk NASH† | | | P-value* |
|---|---|---|---|---|---|
| | | No (n = 371) | Yes (n = 214) | Total (n = 585) | |
| | | Mean (SD)/n (%) | Mean (SD)/n (%) | Mean (SD) / n (%) | |
| | XL | 190 (51.2) | 134 (62.6) | 324 (55.4) | |

†At-risk NASH = Definite NASH on biopsy with a NAFLD Activity Score (NAS) $\geq$ 4 with at least 1 point in each category and fibrosis stage $\geq$ 2.

*Based on chi-square test for categorical data and unequal variance t-test for continuous data.

**LSM: Liver stiffness measurement; CAP: controlled attenuation parameter.

lowest BMI category was 0.94 (95% CI = 0.84, 1.00), and the p-value for difference in AUROCs between the lowest and highest BMI categories was 0.008. No differences were observed by other patient characteristics. There was no difference found by FibroScan® probe type (XL vs. M probe).

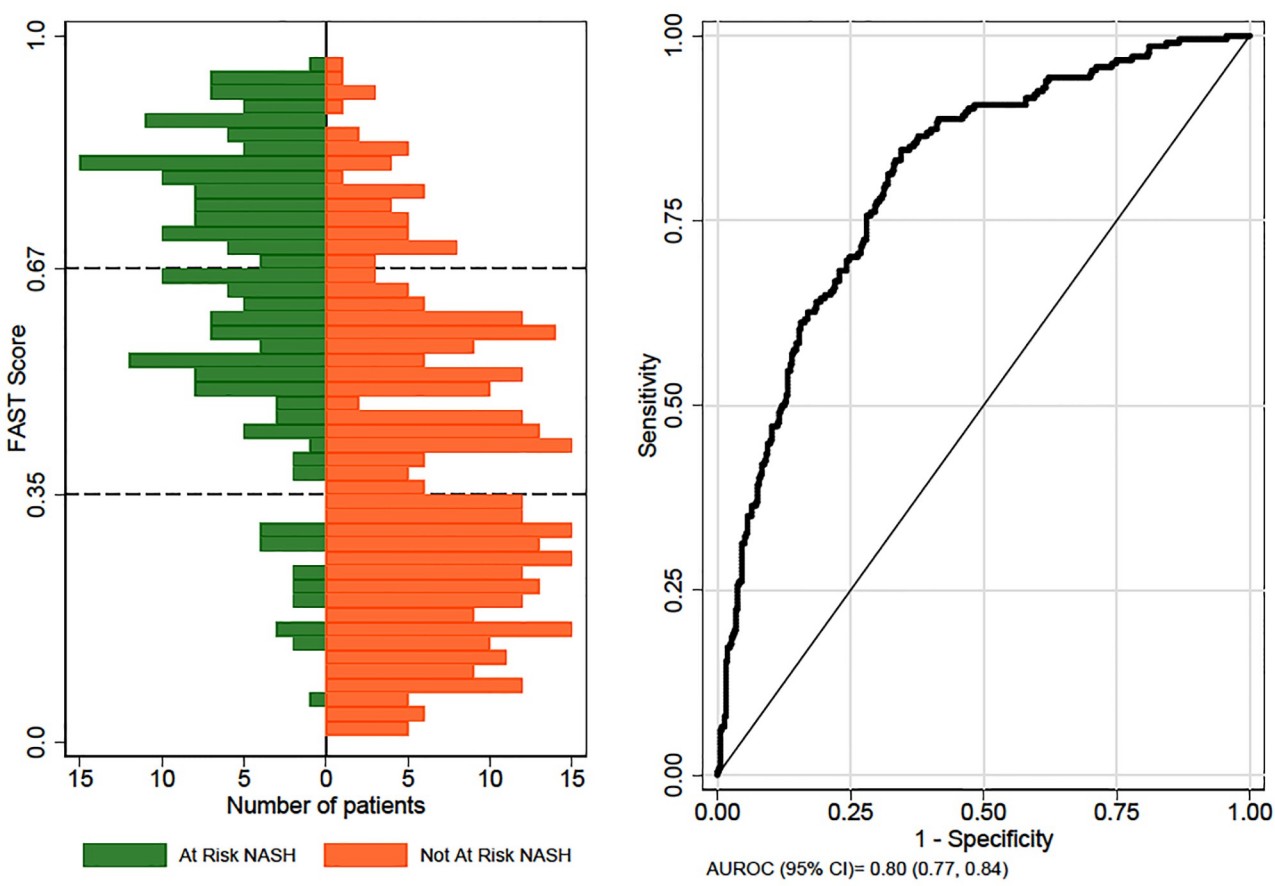

**Fig 1. Diagnostic performance of FAST™ score for detection of at-risk NASH in 585 NAFLD patients.** On the left, histogram showing frequency distribution of FAST™ scores by at-risk NASH status is depicted. Green color depicts scores for patients with at-risk NASH, and orange color shows scores for patients without at-risk NASH. The dotted lines show rule-out cut-off of 0.35 and rule-in cutoff of 0.67. On the right, AUROC of FAST™ for the diagnosis of at-risk NASH is shown.

**Table 2. Diagnostic statistics of FAST™* for at-risk NASH†.**

| Cut-off criteria | N (%)‡ | Cut-off | Sensitivity | Specificity | PPV | NPV |
|---|---|---|---|---|---|---|
| Rule-out zone cutoff (i.e. sensitivity = 0.90) | 208 (35%) | 0.35 | 0.91 | 0.50 | 0.51 | 0.90 |
| Rule-in zone cutoff (i.e., specificity = 0.90) | 160 (27%) | 0.67 | 0.52 | 0.87 | 0.69 | 0.76 |

* FAST™ score = exp(model prediction equation)/ [1 + exp(model prediction equation)], where model prediction equation = -1.65 + 1.07*ln(LSM) + 2.66*$10^{-8}$*(CAP$^3$)– 63.3*(1/AST), where LSM in kPa, CAP in dB/m, and AST in U/L.

†At-risk NASH = Definite NASH + NAS $\geq$4 (steatosis score $\geq$1 and ballooning score $\geq$1 and lobular inflammation score $\geq$1) + fibrosis stage $\geq$2.

‡N = 585 patients with biopsy-confirmed NAFLD at enrollment visit; Prob (at-risk NASH) = 0.37.

There were 20 patients with unreliable LSMs. Although the mean BMI in the 20 patients with unreliable LSMs was significantly higher (mean 37.4 kg/m$^2$) than the 565 patients with reliable LSMs (mean 34.4 kg/m$^2$), the p-value for the comparison of ROC curves by the 4 BMI categories was similar when these 20 patients were excluded (p-value of 0.043 with all patients vs p-value of 0.035 when excluding patients with unreliable LSMs).

**Comparison of diagnostic accuracy of FAST™ vs. other non-invasive diagnostic models for at-risk NASH.** A comparison of the diagnostic accuracy of various non-invasive diagnostic models for at-risk NASH vs. the FAST™ score is shown in Table 5. Analysis of the individual components of the FAST™ score showed that the diagnostic accuracy of each individual component was significantly lower to that of the FAST™ score. The addition of ALT to FAST™ did not result in increase in diagnostic accuracy. AUROC for NFS, FIB-4 index, and APRI were also significantly lower than that of the FAST™ score, with values of 0.67, 0.73, and 0.74, respectively.

## Discussion

In this study, we validated the accuracy of the FAST™ score for identifying patients with at-risk NASH using a large, diverse cohort of U.S. patients with the full spectrum of biopsy-proven NAFLD from the NASH CRN network, a cohort with a prevalence of at-risk NASH of 37%. We found that the FAST™ score has good diagnostic performance characteristics with an AUROC of 0.80. Using a cutoff for sensitivity = 90% of 0.35 and for specificity = 90% of 0.67, the PPV and NPV in our cohort was 0.69 and 0.90, respectively. The performance of FAST™

**Table 3. Diagnostic accuracy of FAST™ at varying prevalences of at-risk NASH.**

| Prevalence of at- risk NASH (p) | Sensitivity = 0.90‡ | | | Specificity = 0.90§ | | |
|---|---|---|---|---|---|---|
| | Specificity | PPV* | NPV† | Sensitivity | PPV* | NPV† |
| Observed in NASH CRN | | | | | | |
| 0.37 | 0.53 | 0.52 | 0.90 | 0.44 | 0.72 | 0.73 |
| Calculated* | | | | | | |
| 0.50 | 0.53 | 0.66 | 0.84 | 0.44 | 0.81 | 0.62 |
| 0.10 | 0.53 | 0.18 | 0.98 | 0.44 | 0.33 | 0.94 |
| 0.05 | 0.53 | 0.09 | 0.99 | 0.44 | 0.19 | 0.97 |

*Positive Predictive Value (PPV) = (p*Sens) / [(p*Sens) + (1-p)*(1-Spec)].

†Negative Predictive Value (NPV) = (1-p)*Spec / [(1-p)*Spec + p*(1-Sens)].

‡FAST score cut-off = 0.38.

§FAST score cut-off = 0.72.

**Table 4. Comparison of diagnostic accuracy of FAST™ for at-risk NASH by subgroups.**

| Subgroup | Category | N | AUROC without cross-validation | P-value |
|---|---|---|---|---|
| **Overall** | | **585** | **0.81** | |
| **Demographic** | | | | |
| Gender | Male | 224 | 0.81 | 0.82 |
| | Female | 361 | 0.81 | |
| Age | < 42 yrs | 146 | 0.82 | 0.74 |
| | 43–60 | 291 | 0.81 | |
| | ≥ 61 yrs | 148 | 0.78 | |
| Race | White | 460 | 0.78 | 0.001 |
| | Non-white | 89 | 0.91 | |
| | Refused | 36 | 0.89 | |
| Ethnicity | Hispanic | 82 | 0.86 | 0.20 |
| | Non-Hispanic | 503 | 0.80 | |
| **Anthropometric** | | | | |
| Body mass index | <30 kg/m$^2$ | 156 | 0.86 | 0.21 |
| | 30–34.9 kg/m$^2$ | 188 | 0.78 | |
| | ≥ 35 kg/m$^2$ | 241 | 0.78 | |
| **Comorbidities** | | | | |
| Diabetes | Yes | 257 | 0.78 | 0.20 |
| | No | 328 | 0.83 | |
| Hyperlipidemia | Yes | 289 | 0.81 | 0.67 |
| | No | 296 | 0.80 | |
| **Study characteristic†** | | | | |
| Probe | XL | 324 | 0.83 | 0.25 |
| | M | 261 | 0.78 | |
| Unreliable LSM | IQR/Med > .3 | 20 | 0.82 | 0.85 |
| | IQR/Med ≤ .3 | 565 | 0.81 | |

†0.5% (3/585) VCTEs had number of exams < 10.

was better in non-whites compared to whites. Additional analysis showed that the diagnostic performance was also better in patients with a normal BMI ($< 25$ kg/m$^2$) compared to those with a a BMI $> 35$ kg/m$^2$.

Our study results are consistent with the findings by Newsome et al., who performed the original study deriving the FAST™ score and found an AUROC of 0.80 in the derivation cohort from England and 0.85 in the pooled validation cohort [15]. Using the dual cutoff approach, the PPV in the derivation cohort, where the prevalence of at-risk NASH was 50%, was 0.83, and the NPV was 0.85. In the seven international external validation cohorts, the NPV was consistently high, ranging from 0.73 to 1.0. The NPV in the pooled external validation cohort, which was comprised of 1026 patients, was 0.94, which was similar to the high NPV of 0.90 that we found in our cohort. There was a greater range in the PPV in the seven external validation cohorts, which ranged from 0.33 to 0.81. The PPV in the pooled cohort was 0.69, which was the same as the PPV in our cohort. The modest PPV found in our cohort where 41% of individuals had a BMI $\geq 40$ kg/m$^2$ reflects that the accuracy of most ultrasound based modalities are restricted in those with severe obesity.

In a study conducted by Oeda et al. in Japan, the AUROC of the FAST™ score for the diagnosis of at-risk NASH was comparable but lower than that reported in our study and the

**Table 5. Comparison of various diagnostic models for at-risk NASH vs. FAST™.**

| Model | AUROC | 95% CI | P-value (Comparison vs FAST™) |
|---|---|---|---|
| **FAST™** | **0.807** | **0.770, 0.843** | -- |
| **FAST™ components** | | | |
| Log LSM | 0.774 | 0.736, 0.811 | 0.04 |
| CAP[3] | 0.564 | 0.515, 0.612 | <0.0001 |
| 1/AST | 0.757 | 0.716, 0.797 | 0.0006 |
| **Hepatic panel components** | | | |
| AST | 0.757 | 0.716, 0.797 | 0.0006 |
| ALT | 0.640 | 0.594, 0.687 | <0.0001 |
| AST/ALT | 0.653 | 0.609, 0.697 | <0.0001 |
| Alkaline phosphatase | 0.608 | 0.560, 0.656 | <0.0001 |
| Total bilirubin | 0.508 | 0.459, 0.556 | <0.0001 |
| Direct bilirubin* | 0.528 | 0.484, 0.572 | <0.0001 |
| Albumin | 0.551 | 0.502, 0.599 | <0.0001 |
| Total protein | 0.534 | 0.484, 0.583 | <0.0001 |
| **Combinations** | | | |
| FAST + ALT | 0.809 | 0.773, 0.845 | 0.23 |
| **Fibrosis models** | | | |
| FIB-4 index | 0.730 | 0.689, 0.771 | 0.0003 |
| NFS | 0.668 | 0.624, 0.712 | <0.0001 |
| APRI | 0.739 | 0.698, 0.780 | <0.0001 |

*There are 8 patients with missing data for direct bilirubin.

†Best subset based on AIC criteria among hepatic panel components.

Newsome et al. study, with a value of 0.76 for both the M and XL probes [20]. As in our study, there was no significant differences in the diagnostic accuracy by probe type. The NPV in the study was high at approximately 0.89. The prevalence of obesity in the overall study population was substantially lower than that of our population at about 25%, which is consistent with the lower prevalence of obesity reported in patients with NAFLD in Asian countries compared to Western countries, even using the definition of obesity in this population of $> 25$ kg/m$^2$ [21]. The prevalence of at-risk NASH was also lower at about 25%.

These results illustrate that the PPV and NPV of the FAST™ score, like for any other tests, are determined by the population to which it is applied [15]. The prevalence of at-risk NASH in the target population will affect the PPV and NPV, and the diagnostic performance will also be determined by the patient mix. The modest PPV, which was found across clinical studies, highlights the utility of the FAST™ score in secondary care settings, where the prevalence of at-risk NASH will be higher than primary care settings, and the PPV will thus increase. The high NPV, which has been found across all existing studies, illustrates the important role that the FAST™ score can serve in identifying patients that are likely to be eligible for enrollment in clinical trials of newer pharmacotherapies for the treatment of NASH. Currently, the majority of NASH clinical trials for non-cirrhotic patients require histological evidence of NASH with Stage 2 or 3 fibrosis as an inclusion criteria [22,23]. The use of the FAST™ score in such settings can thus decrease the number of unnecessary liver biopsies performed and the rate of screening failure for clinical trials, which is currently high.

In clinical practice, a combination of noninvasive tests is often used to estimate the stage of fibrosis in patients with NASH, given the limited diagnostic accuracy of individual tests

compared to the gold standard of liver biopsy. There have been several other prediction models developed for patients with NAFLD based on FibroScan® measurements combined with laboratory tests associated with liver inflammation or fibrosis, such as serum ALT or platelet count. Lee et al. developed the CLA model to discriminate between NASH and simple steatosis, which is based on three independent predictors: CAP > 250 dB/m, LSM > 7 kPa, and ALT > 60 IU/mL [24]. The discriminatory capability was found to be good with an AUROC of 0.812. Okajima et al. devised the LSM/platelet ratio index and found the AUROCs for detecting fibrosis stages ≥ 1, ≥ 2, and ≥ 3 were 0.835, 0.913, and 0.936, respectively, which was greater than that of LSM alone or established scoring symptoms as the NFS, FIB-4 index, and APRI [25]. Jung et al. showed that a magnetic resonance elastography (MRE) score ≥ 3.3 KPa combined with FIB-4 ≥ 1.6 can be used to identify patients with stage 2 fibrosis or higher with a PPV ≥ 90% [26]. Newsome et al. considered five predictor variables in the creation of their model (LSM, CAP, AST, ALT, and AST: ALT ratio) and determined that AST was the best parameter to combine with LSM and CAP [15]. This model had significantly better predictive properties than models with only one or two of these predictors and resulted in the equation for the FAST™ score.

In our study, we similarly found that the diagnostic accuracy of the FAST™ score was significantly greater than that of individuals components of FAST™ as well as the commonly used fibrosis scoring systems NFS, Fibrosis-4 (FIB-4) index, and APRI for the diagnosis of at-risk NASH.

The use of a predictive model based on FibroScan® measurements is valuable as FibroScan® is increasingly used in clinical settings in the U.S. since it received FDA approval in 2013 and has been shown to have good performance characteristics for the detection of advanced fibrosis stage and steatosis grade and low failure rates [12,16,27–30]. A prior study from the NASH CRN showed that, in NAFLD patients from North America, the failure rate of FibroScan® with automatic probe selection for the estimation of LSM and CAP was very low at 3.2% [16]. In addition, the reliability rate was > 95%, and the reproducibility of LSM and CAP measurement was high. Furthermore, FibroScan® is easy to perform, relatively inexpensive compared to liver biopsy or magnetic resonance imaging, and can be performed as a point of care test in clinical practice settings where a FibroScan® device is available. The FAST™ score can be easily calculated using the LSM and CAP measurements along with a serum AST value using a free app from Echosens, myFibroScan, which makes it a convenient tool to use in clinical practice.

The major strengths of this study are the large sample size and multi-center nature of this U.S. based study. Although the original validation study by Newsome et al. included an external U.S. cohort, this was limited to 242 patients at a single center [15]. The original derivation cohort had 350 patients from 7 centers in England, with a mean age of 54 yrs, 43% female, mean BMI 34.2 kg/m$^2$ and 50% with at-risk NASH [15]. Our study population included 585 patients from 8 centers in the U.S., with a mean age of 51 yrs, 62% female, mean BMI 34.5 kg/m2, and 37% with at-risk NASH. We thus studied a patient population with different baseline characteristics including a higher proportion of females and lower proportion with at-risk NASH.

An additional strength of our analysis is that we examined ROC curves by subgroups defined by patient characteristics such as race/ethnicity and BMI which was not perfomed in the original paper. We found that the performance of FAST™ was better in non-whites compared to whites. Our analysis also suggests that FAST™ has better diagnostic performance in patients with a normal BMI compared to morbid obese patients. Race/ethnicity and BMI were also found to be important in a prior NASH CRN study evaluating the performance characteristics of VCTE, as Hispanic ethnicity and BMI category were associated with a higher odds of

unreliable LSM [16]. However, only 20 patients (3.4%) in our cohort had unreliable LSM, which makes it unlikely that this accounted for the differences observed by race and BMI category. Given the high prevalence of obesity in our cohort (73%), the majority of patients (55%) had the XL probe chosen for FibroScan® measurement. We did not find any significant difference between the diagnostic accuracy of FAST™ by probe type, which makes this unlikely to have accounted for differences in diagnostic accuracy according to BMI category. Further studies are needed to confirm differences in the diagnostic accuracy of FAST™ by race/ethnicity and BMI category with detailed information on abdominal fat distribution to explore possible mechanisms to account for such differences.

Of the 61 patients with cirrhosis in our study, only 35 (57%) met the definition of at-risk NASH. Thus, there is a need to develop a non-invasive score to predict the outcome of at-risk NASH or cirrhosis, as cirrhotic patients are at the highest risk of developing complications such as decompensated liver disease and hepatocellular cancer. In addition, there are several ongoing NASH clinical trials which are enrolling cirrhotic patients [31]. Additional models which have high diagnostic accuracy in the identification of patients with NASH who may be eligible for enrollment in clinical trials, including cirrhotic patients, are thus needed.

There are several other strengths of our study. Our study population consisted of a large number of patients seen across 8 U.S. medical centers as part of the NASH CRN Adult DB2 study. All new participants in the Adult Database 2 study required a liver biopsy to be eligible for enrollment. Thus, the decision to proceed with liver biopsy was not based on the results of FibroScan® or laboratory tests, and hence patients with the full spectrum of disease severity were included in the analysis, minimizing selection bias and making the results broadly applicable to patients with imaging evidence of steatosis or having other risk factors for NAFLD. The large sample size increased our ability to detect differences among subgroups in the overall cohort. The study population was relatively diverse, with 14% of patients being of Hispanic ethnicity and 9% being Asian. We examined prediction models for a very important clinical outcome of Stage 2 fibrosis or higher rather than focusing only on more advanced stages of fibrosis. The need for a noninvasive test to identify such patients is critical as the majority of current clinical trials aim to recruit patients with Stage 2 or 3 fibrosis.

The limitations of the study include the absence of longitudinal data on changes in FAST™ score over time in patients with NASH. The requirement of liver biopsy is currently a major barrier to enrollment in NASH clinical trials, and the development of a noninvasive score which can supplant liver biopsy can only take place if longitudinal data confirming the accuracy of a noninvasive score in assessing changes in liver inflammation and fibrosis is available. In addition, the average time between FibroScan® measurement and liver biopsy in the study was 77 days, compared to the shorter time interval of 2 weeks in Newsome et al. study, which may have impacted diagnostic accuracy. The small size of certain subgroups may have also limited diagnostic accuracy in our subgroup analysis. Finally, 38% of patients in the NASH CRN cohort had a FAST™ score that fell into the indeterminate range using the rule-out cutoff of 0.35 and rule-in cutoff of 0.67 for the identification of at-risk NASH. This is similar to the results from the derivation cohort reported by Newsome et al., where 39% of individuals had a FAST™ score in the grey zone between the rule-out and rule-in cutoffs [15]. A recent study found that the combination of MRE and FIB-4 (MEFIB) had a higher diagnostic accuracy and resulted in a smaller percentage of patients in the indeterminate range compared to FAST™ for the detection of stage 2 fibrosis or higher in patients with NAFLD. Further studies are needed to compare such strategies to FAST™ for the detection of at-risk NASH [32].

In summary, this study validated the use of the FAST™ score for the diagnosis of at-risk NASH in a large, diverse cohort of U.S. patients with biopsy-proven NAFLD.

## Supporting information

**S1 Table. % at-risk NASH and FAST™ by fibrosis stage.**
(DOC)

## Acknowledgments

Members of the Nonalcoholic Steatohepatitis Clinical Research Network Adult Clinical Centers

**Cleveland Clinic Foundation, Cleveland, OH**: Daniela Allende, MD; Annette Bellar, MSLA; Jaividhya Dasarathy, MD; Srinivasan Dasarathy, MD; Julie Martridonna; Nicole Welch, MD; Rahul Yerrapothu Arthur J. McCullough, MD (2002–2021)

**Duke University Medical Center, Durham, NC**: Manal F. Abdelmalek, MD, MPH; Mustafa Bashir, MD; Anna Mae Diehl, MD; Cynthia Guy, MD; Christopher Kigongo, MB, CHB; Mariko Kopping, MS, RD; Dawn Piercy, MS, FNP; Naglaa Tawadrous

**Indiana University School of Medicine, Indianapolis, IN**: Timeka Bates, RN; Naga Chalasani, MD; Mandy Cruz, RN; Oscar W. Cummings, MD; Lisa Garrison, RN; Samer Gawrieh, MD; Niharika Samala, MD; Jessie Vaughn, RN; Raj Vuppalanchi, MD

**Saint Louis University, St Louis, MO**: Danielle Carpenter, MD; Theresa Cattoor, RN; Janet Freebersyser, RN; Brent A. Neuschwander-Tetri, MD; Susan Torretta; Elizabeth M. Brunt, MD (2002–2008); Debra King, RN (2004–2015); Jinping Lai, MD (2015–2016); Joan Siegner, RN (2004–2015); Susan Stewart, RN (2004–2015); Kristina Wriston, RN (2015)

**Liver Institute Northwest, Seattle, WA**: Kris V. Kowdley, MD; Luiza Deftu; Tarika Sivakumar; Theresa Dorrian; Heather Harris; My Nguyen

**University of California San Diego, San Diego, CA**: Veeral Ajmera, MD; Cynthia Behling, MD, PhD; Lori Edge; Rohit Loomba, MD, MHSc; Egbert Madamba; Michael S. Middleton, MD, PhD; Lisa Richards, NP; Suzanne Sharpton, MD; Seema Singh; Claude Sirlin, MD

**University of California San Francisco, San Francisco, CA**: Danielle Brandman, MD, MAS; Ryan Gill, MD, PhD; Bilal Hameed, MD; Remilekun Awe; Nathan M. Bass, MD, PhD (2002–2011)

**University of Southern California, Los Angeles, CA**: Daisy Olvera, BA; Norah Terrault, MD, MPH; Liyun Yuan, MD, PhD

**University of Washington Medical Center, Seattle, WA**: Matthew Yeh, MD, PhD

**Virginia Commonwealth University, Richmond, VA**: Amon Asgharpour, MD; Sherry Boyett, RN, BSN; Melissa J. Contos, MD; Velimir AC Luketic, MD; Arun J. Sanyal, MD; Jolene Schlosser, RN, BSN; Mohammad S. Siddiqui, MD

**Washington University, St. Louis, MO**: Elizabeth M. Brunt, MD (2008–2015); Kathryn Fowler, MD (2012–2015)

Resource Centers

**National Cancer Institute, Bethesda, MD**: David E. Kleiner, MD, PhD

**National Institute of Diabetes and Digestive and Kidney Diseases, Bethesda, MD**: Edward C. Doo, MD; Sherry Hall, MS; Jay H. Hoofnagle, MD; Averell H. Sherker, MD; Rebecca Torrance, RN, MS; Patricia R. Robuck, PhD, MPH (2002–2011)

**Data Coordinating Center, Johns Hopkins University, Bloomberg School of Public Health, Baltimore, MD**: Peggy Adamo, BS; Patricia Belt, BS; Jeanne M. Clark, MD, MPH; Jennifer M. DeSanto, RN, BSN, MS; Jill Meinert; Laura Miriel, BS; Emily P. Mitchell, MPH, MBA; Carrie Shade; Jacqueline Smith, AA; Michael Smith, BS; Alice Sternberg, ScM; James Tonascia, PhD; Mark L. Van Natta, MHS; Annette Wagoner; Laura A. Wilson, ScM; Tinsay Woreta,

MD, MPH; Katherine P. Yates, ScM; John Dodge (2002–2018); Michele Donithan, MHS (2002–2017); Milana Isaacson, BS (2002–2018)

## Author Contributions

**Conceptualization:** Mariana Lazo, Arun J. Sanyal.

**Data curation:** Mark L. Van Natta, James Tonascia.

**Formal analysis:** Mark L. Van Natta, James Tonascia.

**Methodology:** Mariana Lazo, Arun J. Sanyal.

**Supervision:** Tinsay A. Woreta, Arun J. Sanyal.

**Validation:** Tinsay A. Woreta, Kris V. Kowdley, James Tonascia, Arun J. Sanyal.

**Visualization:** Tinsay A. Woreta, Mark L. Van Natta, Mariana Lazo, Arunkumar Krishnan, Brent A. Neuschwander-Tetri, Rohit Loomba, Anna Mae Diehl, Manal F. Abdelmalek, Naga Chalasani, Samer Gawrieh, Srinivasan Dasarathy, Raj Vuppalanchi, Mohammad S. Siddiqui, Arthur McCullough, Norah A. Terrault, Cynthia Behling, David E. Kleiner, Mark Fishbein, Paula Hertel, Laura A. Wilson, Emily P. Mitchell, Laura A. Miriel, Jeanne M. Clark, James Tonascia, Arun J. Sanyal.

**Writing – original draft:** Tinsay A. Woreta.

**Writing – review & editing:** Tinsay A. Woreta, Mark L. Van Natta, Mariana Lazo, Arunkumar Krishnan, Brent A. Neuschwander-Tetri, Rohit Loomba, Anna Mae Diehl, Manal F. Abdelmalek, Naga Chalasani, Samer Gawrieh, Srinivasan Dasarathy, Raj Vuppalanchi, Mohammad S. Siddiqui, Kris V. Kowdley, Arthur McCullough, Norah A. Terrault, Cynthia Behling, David E. Kleiner, Mark Fishbein, Paula Hertel, Laura A. Wilson, Emily P. Mitchell, Laura A. Miriel, Jeanne M. Clark, James Tonascia, Arun J. Sanyal.

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
