## [Decision Letter · Decision Letter 0]

31 Jan 2022

PONE-D-22-00397Validation of the accuracy of the FAST™ score for detecting patients with at-risk nonalcoholic steatohepatitis in a North American cohortPLOS ONE

Dear Dr. Woreta,

Thank you for submitting your manuscript to PLOS ONE. After careful consideration, we feel that it has merit but does not fully meet PLOS ONE’s publication criteria as it currently stands. Therefore, we invite you to submit a revised version of the manuscript that addresses the points raised during the review process. As you can see, both reviewers appreciated your work and only relatively minor changes are needed.

We look forward to receiving your revised manuscript.

Kind regards,

Pavel Strnad

Academic Editor

PLOS ONE

Journal Requirements:

"The Nonalcoholic Steatohepatitis Clinical Research Network (NASH CRN) is supported by the National Institute of Diabetes and Digestive and Kidney Diseases (NIDDK) (grants U01DK061713, U01DK061718, U01DK061728, U01DK061731, U01DK061732, U01DK061734, U01DK061737, U01DK061738, U01DK061730, U24DK061730). Additional support is received from the National Center for Advancing Translational Sciences (NCATS) (grants UL1TR000439, UL1TR000436, UL1TR000006, UL1TR000448, UL1TR000100, UL1TR000004, UL1TR000423, UL1TR002649). This research was supported in part by the Intramural Research Program of the NIH, National Cancer Institute. All FibroScan® 502 Touch systems were provided to the NASH CRN Investigators by Echosens™ through a Clinical Trial Agreement with the NIDDK."

"Dr. Manal Abdelmalek: None for this project. Dr. Abdelmalek has served as a consultant to Merck, Bristol Myers Squibb,  Novartis, Novo Nordisk, Hanmi, TaiwanJ , Madrigal and NGM Bio.  Her institution has received grant support from Allergan, Intercept, Boehringer-Ingelheim, Bristol Myers Squibb, Madrigal, Novo Nordisk, NGM Bio, Novartis, Viking, Hanmi, TARGET NASH, Celgene, and Genentech.  She receives royalties from Elsevier and Up-to-Date

Dr. Cynthia Behling: Dr. Behling is a consultant for ICON, COVANCE, Eli Lilly, Hologic, and Leica outside the submitted work.

Dr. Naga Chalasani: There are none for this paper.  For full disclosure, Dr. Chalasani has ongoing consulting activities (or had in preceding 12 months) with Abbvie, Madrigal, Zydus, Galectin, Boehringer-Ingelheim, Altimmune, and Foresite.  These consulting activities are generally in the areas of nonalcoholic fatty liver disease and drug hepatotoxicity. Dr. Chalasani has equity in RestUp, a home health care provider agency. Dr. Chalasani receives research grant support from DSM and Exact Sciences where his institution receives the funding. 

Dr. Anna Mae Diehl: Dr. Diehl has consulted for Allergan, Alderya Therapeutics, Boehringer-Ingelheim, Celgene, Filcitrine, IBM Watson Health, Lumena, Merk, Novartis, Pfizer, Pliant, Roche, Quest Diagnostics, and twoXAR. She has research collaborations with Allergan, Boehringer-Ingelheim, Bristol Myers Squibb, Conatus, Exalenze, Galactin, Galmed, Genfit, Gilead, Hanmi, Hi La, Immuron, Intercept, Madrigal Metabolomics, NGM Pharmaceuticals, Prometheus, and Shire.

Dr. Samer Gawrieh: Dr. Gawrieh has consulted for TransMedics and Pfizer and received research grant support from Cirius, Galmed, Viking, Zydus and Sonic Incytes.

Dr. Kris Kowdley: Dr. Kowdley has consulted for Akero, Calliditas, Corcept, CymaBay, Enanta, Genfit, Gilead,HighTide, Inipharm and Intercept. His institution has received grant and research support from Allergan, Enanta, Galectin, Gilead, Immuron, Intercept, Novartis, Prometheus, and Zydus. Dr. Kowdley is on the Advisory Board for Conatus and Gilead, and is on the Speaker Bureau for Abbvie, Gilead and Intercept.

Dr. Rohit Loomba: There are none for this paper. Dr. Loomba serves as a consultant or advisory board member for 89bio, Alnylam, Arrowhead Pharmaceuticals, AstraZeneca, Boehringer Ingelheim, Bristol-Myer Squibb, Cirius, CohBar, DiCerna, Galmed, Gilead, Glympse bio, Intercept, Ionis, Metacrine, NGM Biopharmaceuticals, Novo Nordisk, Pfizer, Sagimet and Viking Therapeutics. In addition, his institution has received grant support from Allergan, Boehringer-Ingelheim, Bristol-Myers Squibb, Eli Lilly and Company, Galmed Pharmaceuticals, Genfit, Gilead, Intercept, Inventiva, Janssen, Madrigal Pharmaceuticals, NGM Biopharmaceuticals, Novartis, Pfizer, pH Pharma, and Siemens.  He is also co-founder of Liponexus, Inc. Dr. Loomba receives funding support from NIEHS (5P42ES010337), NCATS (5UL1TR001442), NIDDK (R01DK106419, 1R01DK121378, R01 DK124318, P30DK120515), and DOD PRCRP (CA170674P2). 

Dr. Brent Neuschwander-Tetri: None for the project. Consultant or advisor for Akero, Alimentiv, Allergan, Alnylam, Amgen, Arrowhead,  Axcella, Boehringer Ingelheim, BMS, Durect, Enanta, Fortress, Gelesis, Genfit, Gilead, HepGene, High Tide, HistoIndex, Intercept, Ionis, LG Chem, Lipocine, Madrigal, Medimmune, Merck, Mirum, NGM, NovoNordisk, pH-Pharma, Sagimet, Siemens, Theratechnologies, 89Bio; Institutional research grants: Allergan, BMS, Cirius, Enanta, Genfit, Gilead, Intercept, Madrigal, NGM

Dr. Arun Sanyal: None for this project. Dr. Sanyal is President of Sanyal Biotechnology and has stock options in Genfit, Akarna, Tiziana, Indalo, Durect Inversago and Galmed. He has served as a consultant to Astra Zeneca, Nitto Denko, Conatus, Nimbus, Salix, Tobira, Takeda, Jannsen, Gilead, Terns, Birdrock, Merck, Valeant, Boehringer-Ingelheim, Bristol Myers Squibb, Lilly, Hemoshear, Zafgen, Novartis, Novo Nordisk, Pfizer, Exhalenz and Genfit. He has been an unpaid consultant to Intercept, Echosens, Immuron, Galectin, Fractyl, Syntlogic, Affimune, Chemomab, Zydus, Nordic Bioscience, Albireo, Prosciento, Surrozen. His institution has received grant support from Gilead, Salix, Tobira, Bristol Myers, Shire, Intercept, Merck, Astra Zeneca, Malinckrodt, Cumberland and Novartis. He receives royalties from Elsevier and UptoDate.

Dr. Mohammad Siddiqui: Dr. Siddiqui is a consultant for Pfizer and is on an advising board for AMRA.

Dr. Norah Terrault: Dr. Terrault received institutional grant support from Gilead Sciences, Roche-Genentech and GSK, and consulting for Gilead Sciences, Saol Therapeutics and Moderna.

Dr. Raj Vuppalanchi: Dr. Vuppalanchi received a one-time consultant fee from EchoSens on spleen stiffness measurement.

No conflicts of interest: Jeanne Clark, Srinivasan Dasarathy, Mark Fishbein, Paula Hertel, David Kleiner, Arunkumar Krishnan, Mariana Lazo, Arthur McCullough, Laura Miriel, Emily Mitchell, James Tonascia, Mark Van Natta, Laura Wilson, Tinsay Woreta"

6. One of the noted authors is a group or consortium NASH Clincial Research Network. In addition to naming the author group, please list the individual authors and affiliations within this group in the acknowledgments section of your manuscript. Please also indicate clearly a lead author for this group along with a contact email address.

Reviewers' comments:

Reviewer's Responses to Questions

**Comments to the Author**

1. Is the manuscript technically sound, and do the data support the conclusions?

Reviewer #1: Yes

Reviewer #2: Yes

2. Has the statistical analysis been performed appropriately and rigorously? 

Reviewer #1: Yes

Reviewer #2: Yes

3. Have the authors made all data underlying the findings in their manuscript fully available?

Reviewer #1: Yes

Reviewer #2: Yes

4. Is the manuscript presented in an intelligible fashion and written in standard English?

Reviewer #1: Yes

Reviewer #2: Yes

5. Review Comments to the Author

Reviewer #1: The article by Woreta TA et al entitled ‘ Validation of the accuracy of the FAST™ score for detecting patients with at-risk nonalcoholic steatohepatitis in a North American cohort ‘ describes the validation of the non-invasive FASTscore to stratify patients with non-alcoholic fatty liver disease based on the presence of the histological features NAS>=4 and fibrosis. The paper uses a well-characterised cohort of 585 US patients with biopsy-proven NAFLD enrolled in the NASH Clinical Research Network (NASH CRN) Adult Database (D2) cohort study. The paper validates the previously reported FAST score (Newsome P et al 2020 Lancet Gastro&Hep) with similar rule-in and rule-out cut-offs. Patients in this study were enrolled based on the availability of FibroScan with LSM and CAP, aminotransferase levels and a centrally read liver biopsy within a timeframe of 6 months. The paper is well written and timely, and provides interesting insights in the performance of the score in an independent cohort. Only a few minor comments came to mind which are described below.

Minor comments

The description of the ‘derivation cohort’ in the Material & Methods is confusing and gives the impression that the 350 patients are part of the entire study cohort of 585 patients. The data and results of the derivation cohort have already been published. If the authors aim to perform a kind of meta-analysis comparing different cohorts then this should be clearly stated and explained in detail. If the aim is to validate an already published/established score, then the authors should focus on the new cohort and leave the description of a previously published cohort out of the M&M and results.

The nomenclature of ‘derivation cohort’ in the context of validating an established non-invasive score is rather confusing as nothing is being derived in this manuscript.

The abbreviation NAFLD is used for the ‘umbrella’ term non-alcoholic fatty liver disease as well as for non-alcoholic fatty liver (NAFL). I would recommend to use NAFLD and NAFL.

The definition used for at risk NASH and main outcome for this study changes slightly throughout the manuscript. In the abstract and p11, the authors define it as ‘NAS of 4 or higher and fibrosis stage of 2 or higher’. On p13, it is defined as ‘NAS 4 or higher with at least 1 point in each category and a fibrosis stage 2 or higher’, so NASH + NAS>=4 + F>=2. Presumably the latter definition has been used. This should be clarified and adjusted where necessary.

Not clear which statistical test was implanted to compare ROC curves? DeLong test? This should be clearly described in the M&M.

Reviewer #2: I’ve read with great interest the manuscript “Validation of the accuracy of the FAST score for detecting patient with at risk nonalcoholic steatohepatititis in a North American Cohort”. This study represents a validation study in which the FAST score was validated in a multi-racial North American NASH population, and confirms the diagnostic performance in this well-defined NAFLD cohort, as well as in several subgroups based on weigth and race.

The study is relevant, since good performing non-invasive tests are warranted for detection at risk NASH patients as well as inclusion for clinical studies. The methodology and statistics used are adequate and presented in detail, and the conclusion is presentered adequately.

However, since the original validation study (ref 15 Newsome et al) also included an external US NAFLD population in which the FAST score was validated, my main question is why the authors expected that the diagnostic performance of the FAST would be different in this specific NAFLD cohort? Were there large differences in baseline characteristics between the cohort in this study and the original validation study? In case the cohorts were similar, what is the added value of this study (larger population? Subgroups)? Please specifiy.

I also have some minor questions.

- Was the fibroscan performed standard in this cohort or by indication?

- How did the Pathology committee deal with discrepancies between histologic scores between observers?

- The authors mention that there was no difference found by Fibroscan probe type (XL vs M) when probe type was selected by the machine, was there a difference when the operator chose the probe type?

- The diagnostic accuracy of the CAP seems low when looking at tabel 5, what is the added value of this parameter when added tot he FAST-score?

- There were only 20 patients with unreliable LSM, what were the characteristics? Were they all morbid obese? Were these results also included in the analyses, and did the authors perform a sensitivity analysis without these unreliable results to check whether there is still a significiant difference between BMI groups?

6. PLOS authors have the option to publish the peer review history of their article (what does this mean?). If published, this will include your full peer review and any attached files.

Reviewer #1: No

Reviewer #2: No

---

## [Author Response · Author response to Decision Letter 0]

18 Mar 2022

Editor Comments:

Answer: We have edited our manuscript to ensure it meets PLOS ONE's style requirements.

2. Please review your reference list to ensure that it is complete and correct. 

Answer: We have reviewed our reference list to ensure it is complete and correct. We didn't include any retracted references in our manuscript. 

3. We note that the grant information you provided in the 'Funding Information' and 'Financial Disclosure' sections do not match. 

Answer: We have reviewed our grant information and have confirmed these are the correct grant numbers.

"The Nonalcoholic Steatohepatitis Clinical Research Network (NASH CRN) is supported by the National Institute of Diabetes and Digestive and Kidney Diseases (NIDDK) (grants U01DK061713, U01DK061718, U01DK061728, U01DK061731, U01DK061732, U01DK061734, U01DK061737, U01DK061738, U01DK061730, U24DK061730). Additional support is received from the National Center for Advancing Translational Sciences (NCATS) (grants UL1TR000439, UL1TR000436, UL1TR000006, UL1TR000448, UL1TR000100, UL1TR000004, UL1TR000423, UL1TR002649). This research was supported in part by the Intramural Research Program of the NIH, National Cancer Institute. All FibroScan® 502 Touch systems were provided to the NASH CRN Investigators by Echosens™ through a Clinical Trial Agreement with the NIDDK." 

Answer: We have added the statement "The funders had no role in study design, data collection and analysis, decision to publish, or preparation of the manuscript." in our cover letter. 

"Dr. Manal Abdelmalek: None for this project. Dr. Abdelmalek has served as a consultant to Merck, Bristol Myers Squibb, Novartis, Novo Nordisk, Hanmi, TaiwanJ , Madrigal and NGM Bio. Her institution has received grant support from Allergan, Intercept, Boehringer-Ingelheim, Bristol Myers Squibb, Madrigal, Novo Nordisk, NGM Bio, Novartis, Viking, Hanmi, TARGET NASH, Celgene, and Genentech. She receives royalties from Elsevier and Up-to-Date

Dr. Cynthia Behling: Dr. Behling is a consultant for ICON, COVANCE, Eli Lilly, Hologic, and Leica outside the submitted work.

Dr. Naga Chalasani: There are none for this paper. For full disclosure, Dr. Chalasani has ongoing consulting activities (or had in preceding 12 months) with Abbvie, Madrigal, Zydus, Galectin, Boehringer-Ingelheim, Altimmune, and Foresite. These consulting activities are generally in the areas of nonalcoholic fatty liver disease and drug hepatotoxicity. Dr. Chalasani has equity in RestUp, a home health care provider agency. Dr. Chalasani receives research grant support from DSM and Exact Sciences where his institution receives the funding. 

Dr. Anna Mae Diehl: Dr. Diehl has consulted for Allergan, Alderya Therapeutics, Boehringer-Ingelheim, Celgene, Filcitrine, IBM Watson Health, Lumena, Merk, Novartis, Pfizer, Pliant, Roche, Quest Diagnostics, and twoXAR. She has research collaborations with Allergan, Boehringer-Ingelheim, Bristol Myers Squibb, Conatus, Exalenze, Galactin, Galmed, Genfit, Gilead, Hanmi, Hi La, Immuron, Intercept, Madrigal Metabolomics, NGM Pharmaceuticals, Prometheus, and Shire.

Dr. Samer Gawrieh: Dr. Gawrieh has consulted for TransMedics and Pfizer and received research grant support from Cirius, Galmed, Viking, Zydus and Sonic Incytes.

Dr. Kris Kowdley: Dr. Kowdley has consulted for Akero, Calliditas, Corcept, CymaBay, Enanta, Genfit, Gilead,HighTide, Inipharm and Intercept. His institution has received grant and research support from Allergan, Enanta, Galectin, Gilead, Immuron, Intercept, Novartis, Prometheus, and Zydus. Dr. Kowdley is on the Advisory Board for Conatus and Gilead, and is on the Speaker Bureau for Abbvie, Gilead and Intercept.

Dr. Rohit Loomba: There are none for this paper. Dr. Loomba serves as a consultant or advisory board member for 89bio, Alnylam, Arrowhead Pharmaceuticals, AstraZeneca, Boehringer Ingelheim, Bristol-Myer Squibb, Cirius, CohBar, DiCerna, Galmed, Gilead, Glympse bio, Intercept, Ionis, Metacrine, NGM Biopharmaceuticals, Novo Nordisk, Pfizer, Sagimet and Viking Therapeutics. In addition, his institution has received grant support from Allergan, Boehringer-Ingelheim, Bristol-Myers Squibb, Eli Lilly and Company, Galmed Pharmaceuticals, Genfit, Gilead, Intercept, Inventiva, Janssen, Madrigal Pharmaceuticals, NGM Biopharmaceuticals, Novartis, Pfizer, pH Pharma, and Siemens. He is also co-founder of Liponexus, Inc. Dr. Loomba receives funding support from NIEHS (5P42ES010337), NCATS (5UL1TR001442), NIDDK (R01DK106419, 1R01DK121378, R01 DK124318, P30DK120515), and DOD PRCRP (CA170674P2). 

Dr. Brent Neuschwander-Tetri: None for the project. Consultant or advisor for Akero, Alimentiv, Allergan, Alnylam, Amgen, Arrowhead, Axcella, Boehringer Ingelheim, BMS, Durect, Enanta, Fortress, Gelesis, Genfit, Gilead, HepGene, High Tide, HistoIndex, Intercept, Ionis, LG Chem, Lipocine, Madrigal, Medimmune, Merck, Mirum, NGM, NovoNordisk, pH-Pharma, Sagimet, Siemens, Theratechnologies, 89Bio; Institutional research grants: Allergan, BMS, Cirius, Enanta, Genfit, Gilead, Intercept, Madrigal, NGM

Dr. Arun Sanyal: None for this project. Dr. Sanyal is President of Sanyal Biotechnology and has stock options in Genfit, Akarna, Tiziana, Indalo, Durect Inversago and Galmed. He has served as a consultant to Astra Zeneca, Nitto Denko, Conatus, Nimbus, Salix, Tobira, Takeda, Jannsen, Gilead, Terns, Birdrock, Merck, Valeant, Boehringer-Ingelheim, Bristol Myers Squibb, Lilly, Hemoshear, Zafgen, Novartis, Novo Nordisk, Pfizer, Exhalenz and Genfit. He has been an unpaid consultant to Intercept, Echosens, Immuron, Galectin, Fractyl, Syntlogic, Affimune, Chemomab, Zydus, Nordic Bioscience, Albireo, Prosciento, Surrozen. His institution has received grant support from Gilead, Salix, Tobira, Bristol Myers, Shire, Intercept, Merck, Astra Zeneca, Malinckrodt, Cumberland and Novartis. He receives royalties from Elsevier and UptoDate.

Dr. Mohammad Siddiqui: Dr. Siddiqui is a consultant for Pfizer and is on an advising board for AMRA.

Dr. Norah Terrault: Dr. Terrault received institutional grant support from Gilead Sciences, Roche-Genentech and GSK, and consulting for Gilead Sciences, Saol Therapeutics and Moderna.

Dr. Raj Vuppalanchi: Dr. Vuppalanchi received a one-time consultant fee from EchoSens on spleen stiffness measurement.

No conflicts of interest: Jeanne Clark, Srinivasan Dasarathy, Mark Fishbein, Paula Hertel, David Kleiner, Arunkumar Krishnan, Mariana Lazo, Arthur McCullough, Laura Miriel, Emily Mitchell, James Tonascia, Mark Van Natta, Laura Wilson, Tinsay Woreta"

Please confirm that this does not alter your adherence to all PLOS ONE policies on sharing data and materials, by including the following statement: "This does not alter our adherence to PLOS ONE policies on sharing data and materials." (as detailed online in our guide for authors http://journals.plos.org/plosone/s/competing-interests). If there are restrictions on sharing of data and/or materials, please state these. Please note that we cannot proceed with consideration of your article until this information has been declared. 

Answer: We have added the statement "This does not alter our adherence to PLOS ONE policies on sharing data and materials." in our cover letter.

6. One of the noted authors is a group or consortium NASH Clincial Research Network. In addition to naming the author group, please list the individual authors and affiliations within this group in the acknowledgments section of your manuscript. Please also indicate clearly a lead author for this group along with a contact email address.

Answer: We have added the NASH CRN and the individual authors and affiliations to the acknowledgements section.

Reviewer #1

Minor comments

1. The description of the 'derivation cohort' in the Material & Methods is confusing and gives the impression that the 350 patients are part of the entire study cohort of 585 patients. The data and results of the derivation cohort have already been published. If the authors aim to perform a kind of meta-analysis comparing different cohorts then this should be clearly stated and explained in detail. If the aim is to validate an already published/established score, then the authors should focus on the new cohort and leave the description of a previously published cohort out of the M&M and results. The nomenclature of 'derivation cohort' in the context of validating an established non-invasive score is rather confusing as nothing is being derived in this manuscript.

Answer: We agree and have removed the section about the derivation cohort from the Materials & Methods and Results section.

2. The abbreviation NAFLD is used for the 'umbrella' term non-alcoholic fatty liver disease as well as for non-alcoholic fatty liver (NAFL). I would recommend to use NAFLD and NAFL.

Answer: We used the term NAFLD to be consistent with our prior NASH CRN publications. We did not characterize any patients with NAFL alone so did not use this term.

3. The definition used for at risk NASH and main outcome for this study changes slightly throughout the manuscript. In the abstract and p11, the authors define it as 'NAS of 4 or higher and fibrosis stage of 2 or higher'. On p13, it is defined as 'NAS 4 or higher with at least 1 point in each category and a fibrosis stage 2 or higher', so NASH + NAS>=4 + F>=2. Presumably the latter definition has been used. This should be clarified and adjusted where necessary.

Answer: We have clarified that at-risk NASH is defined as NAS score 4 or higher with at least point in each category and a fibrosis stage 2 or higher.

4. Not clear which statistical test was implanted to compare ROC curves? DeLong test? This should be clearly described in the M&M.

Answer: We used the Delong test to compare ROC curves and added this statement and reference to the M&M section.

Reviewer #2: 

1. I've read with great interest the manuscript "Validation of the accuracy of the FAST score for detecting patient with at risk nonalcoholic steatohepatititis in a North American Cohort". This study represents a validation study in which the FAST score was validated in a multi-racial North American NASH population, and confirms the diagnostic performance in this well-defined NAFLD cohort, as well as in several subgroups based on weigth and race.

The study is relevant, since good performing non-invasive tests are warranted for detection at risk NASH patients as well as inclusion for clinical studies. The methodology and statistics used are adequate and presented in detail, and the conclusion is presented adequately.

However, since the original validation study (ref 15 Newsome et al) also included an external US NAFLD population in which the FAST score was validated, my main question is why the authors expected that the diagnostic performance of the FAST would be different in this specific NAFLD cohort? Were there large differences in baseline characteristics between the cohort in this study and the original validation study? In case the cohorts were similar, what is the added value of this study (larger population? Subgroups)? Please specifiy.

Answer: The external USA NAFLD population in the Newsome study was in 242 patients at a single center. The original derivation cohort had 350 patients from 7 centers in England, mean age=54 yrs, 43% female, mean BMI=34.2 kg/m2 and 50% with at risk NASH. Our population included 585 patients from 8 centers in the USA, mean age 51 yrs, 62% female, mean BMI=34.5 kg/m2 and 37% with at risk NASH. The rationale for this validation analysis was the added value using a multi-center USA cohort, additional patients with different baseline characteristics including higher % female and lower proportion with at-risk NASH. Another main focus of the analysis was to examine ROC curves by subgroups which was not done in the original paper and compare the FAST score for diagnosing at risk NASH with other common non-invasive liver tests.

We added these points to the discussion to further elaborate.

Minor questions:

1.Was the fibroscan performed standard in this cohort or by indication?

Answer: Fibroscan was performed standard in this cohort and not by indication. We added a statement in the methods section to clarify.

2.How did the Pathology committee deal with discrepancies between histologic scores between observers?

- The authors mention that there was no difference found by Fibroscan probe type (XL vs M) when probe type was selected by the machine, was there a difference when the operator chose the probe type?

Answer: We did not collect data on whether the machine picked the probe type or the operator picked the probe type, only which probe type was used. Operators were trained and certified and followed a written protocol which stipulated when to use the appropriate probe type. We thus removed the statement regarding probe type being selected by machine.

3. The diagnostic accuracy of the CAP seems low when looking at table 5, what is the added value of this parameter when added to the FAST-score?

Answer: CAP is already included as a parameter in the FAST score (see formula).

4. There were only 20 patients with unreliable LSM, what were the characteristics? Were they all morbid obese? Were these results also included in the analyses, and did the authors perform a sensitivity analysis without these unreliable results to check whether there is still a significant difference between BMI groups?

Answer: Although the mean BMI in the 20 patients with unreliable LSMs was significantly (t-test p=0.04) higher mean=37.4 kg/m2 vs the 565 patients with reliable LSMs mean=34.4, the p-value for comparison of ROC curves by the 4 BMI categories was similar when these 20 patients were excluded (p=0.043 with all patients vs p=0.035 excl patients with unreliable LSMs). We added a statement in the results section regarding this sensitivity analysis.

---

## [Decision Letter · Decision Letter 1]

29 Mar 2022

Validation of the accuracy of the FASTTM score for detecting patients with at-risk nonalcoholic steatohepatitis (NASH) in a North American cohort and comparison to other non-invasive algorithms

PONE-D-22-00397R1

Dear Dr. Woreta,

We’re pleased to inform you that your manuscript has been judged scientifically suitable for publication and will be formally accepted for publication once it meets all outstanding technical requirements.

Kind regards,

Pavel Strnad

Academic Editor

PLOS ONE

Additional Editor Comments (optional):

Reviewers' comments:

Reviewer's Responses to Questions

**Comments to the Author**

1. If the authors have adequately addressed your comments raised in a previous round of review and you feel that this manuscript is now acceptable for publication, you may indicate that here to bypass the “Comments to the Author” section, enter your conflict of interest statement in the “Confidential to Editor” section, and submit your "Accept" recommendation.

Reviewer #1: All comments have been addressed

2. Is the manuscript technically sound, and do the data support the conclusions?

Reviewer #1: Yes

3. Has the statistical analysis been performed appropriately and rigorously? 

Reviewer #1: Yes

4. Have the authors made all data underlying the findings in their manuscript fully available?

Reviewer #1: Yes

5. Is the manuscript presented in an intelligible fashion and written in standard English?

Reviewer #1: Yes

6. Review Comments to the Author

Reviewer #1: The authors have adequately addressed my comments.

7. PLOS authors have the option to publish the peer review history of their article (what does this mean?). If published, this will include your full peer review and any attached files.

Reviewer #1: No

---

## [Editor Report · Acceptance letter]

8 Apr 2022

PONE-D-22-00397R1 

Validation of the accuracy of the FAST score for detecting patients with at-risk nonalcoholic steatohepatitis (NASH) in a North American cohort and comparison to other non-invasive algorithms 

Dear Dr. Woreta:

I'm pleased to inform you that your manuscript has been deemed suitable for publication in PLOS ONE. Congratulations! Your manuscript is now with our production department. 

Kind regards, 

on behalf of

Dr. Pavel Strnad 

Academic Editor

PLOS ONE